# Ankle Doppler for Cuffless Ankle Brachial Index Estimation and Peripheral Artery Disease Diagnosis Independent of Diabetes

**DOI:** 10.3390/jcm12010097

**Published:** 2022-12-22

**Authors:** Alexander D. Rodway, Darren Cheal, Charlotte Allan, Felipe Pazos-Casal, Lydia Hanna, Benjamin C. T. Field, Ajay Pankhania, Philip J. Aston, Simon S. Skene, Gary D. Maytham, Christian Heiss

**Affiliations:** 1Surrey and Sussex Healthcare NHS Trust, East Surrey Hospital, Redhill RH1 5RH, UK; 2Brighton and Sussex University Hospitals NHS Trust, Brighton BN2 5BE, UK; 3Department of Surgery and Cancer, Imperial College London, London SW7 2BX, UK; 4Department of Clinical and Experimental Medicine, Faculty of Health and Medical Sciences, University of Surrey, Guildford GU2 7XH, UK; 5Department of Mathematics, University of Surrey, Guildford GU2 7XH, UK; 6St George’s University Hospitals NHS Foundation Trust, London SW17 0QT, UK

**Keywords:** ankle brachial pressure index, Doppler waveform, peripheral artery disease

## Abstract

Ankle brachial pressure index (ABPI) is the first-line test to diagnose peripheral artery disease (PAD). Its adoption in clinical practice is poor and its validity, particularly in diabetes, is limited. We hypothesised that ABPI can be accurately and precisely estimated based on cuffless Doppler waveforms. Retrospective analysis of standard ABPI and handheld Doppler waveform characteristics (*n* = 200). Prospective analysis of angle-corrected Doppler acceleration index (AccI, *n* = 148) and standard ABPI with testing of performance to diagnose PAD as assessed with imaging reference standards in consecutive patients. The highest AccI from handheld Doppler at ankle arteries was significantly logarithmically associated with the highest standard ABPI (E[y] = 0.32 ln [1.71 ∗ x + 1], *p* < 0.001, *R*^2^ = 0.68, *n* = 100 limbs). Estimated ABPI (eABPI) based on AccI closely resembled ABPI (r = 0.81, *p* < 0.001, average deviation −0.01 ± 0.13 [SD], *n* = 100 limbs). AccI from angle-corrected Doppler in patients without overt media sclerosis (ABPI ≤ 1.1) improved ABPI prediction (E[y] = 0.297 ∗ ln[0.039 ∗ x + 1], *R*^2^ = 0.92, *p* = 0.006, average deviation 0.00 ± 0.08, *n* = 100). In a population (*n* = 148 limbs) including diabetes (56%), chronic limb-threatening ischaemia (51%) and media sclerosis (32%), receiver operating characteristics analysis of (angle-corrected) eABPI performed significantly better than standard ABPI to diagnose PAD defined by ultrasound (ROC AUC = 0.99 ± 0.01, *p* < 0.001; sensitivity: 97%, specificity: 96%) at the ≤0.9 cut-off. This was confirmed with CT angiography (ROC AUC = 0.98, *p* < 0.001, sensitivity: 97%, specificity: 100%) and was independent of the presence of diabetes (*p* = 0.608). ABPI can be estimated based on ankle Doppler AccI without compression, and eABPI performs better than standard ABPI to diagnose PAD independent of diabetes. eABPI has the potential to be included as a standard component of lower extremity ultrasound.

## 1. Introduction

Peripheral artery disease (PAD) is a growing healthcare burden worldwide that is underdiagnosed and undertreated and may affect more than 20% of the elderly population [1,2,3]. Up to 50% of patients with diabetes may have PAD [4]. PAD not only leads to leg-specific symptoms and amputations that impair the quality of life, but it is a strong predictor of poly-vascular disease [5], and mortality is comparable to common cancers [6,7]. Early diagnosis of PAD could improve health outcomes by allowing early interventions to manage cardiovascular risk factors and prevent the functional decline of the lower extremities. In patients with tissue loss, the diagnosis of PAD is critical to initiate early revascularisation and prevent amputations.

Currently, the ankle brachial pressure index (ABPI) is the first-line test to diagnose PAD and is an important biomarker of cardiovascular risk [8,9]. However, its adoption in primary healthcare settings is limited, and its validity, particularly in patients with diabetes and media sclerosis, has been questioned [9]. Media sclerosis may render arteries incompressible, resulting in an overestimation of ankle pressures by ABPI, i.e., misleadingly normal ABPI values. Confirming this, a recent meta-analysis showed that pooled estimates for ABPI detecting ≥50% stenosis, based on imaging reference, showed a sensitivity of only 61% (95% CI: 55–69), with a specificity of 92% (95% CI: 89–95) [10].

Another approach to detecting impaired lower extremity perfusion is to analyse arterial Doppler waveform spectra at the ankle. It has long been recognised that the characteristic tri- or multiphasic shape is lost distal to a stenosis or occlusion [11]. Depending on the degree of perfusion deficit, the Doppler waveform becomes monophasic, the systolic upstroke flattens, and diastolic flow increases, leading to *pulsus tardus et parvus*. The more recent literature indicates that Doppler waveform analysis may be useful for detecting severe PAD without requiring compression of arteries [12].

Here, we hypothesised that the ABPI can be estimated based on the Doppler spectrum recorded at the ankle. In the current manuscript, we validate the estimation of ABPI (eABPI) based on the Doppler Acceleration Index (AccI), providing a simple transfer function that can be applied to any Doppler signal obtained at the ankle arteries without requiring compression with a cuff.

## 2. Materials and Methods

We analysed 3 datasets. The first two anonymised datasets each consisted of *n* = 100 leg measurements on consecutive patients without diabetes attending the Vascular Ultrasound Laboratory at the Royal Sussex University Hospital, Brighton, UK. The available data were standard ABPI values and handheld Doppler waveform characteristics (Peak Systolic Amplitude, Rise Time, and AccI, defined as the slope of the line connecting the foot and peak of the systolic upstroke) taken at the ankle. All measurements were performed by one of the authors (DC). The third dataset (*n* = 148 legs, *n* = 85 consecutive patients) was part of a prospective audit performed by the vascular team at East Surrey Hospital, Surrey and Sussex Healthcare NHS Trust, Redhill, UK (January–May 2022). Patient data were included if measurement of angle-corrected AccI in the dorsalis pedis artery and posterior tibial artery at the ankle was accompanied by a full vascular ultrasound to investigate the presence or absence of significant PAD as part of routine clinical practice. Vascular ultrasound exams were performed by qualified and experienced sonographers in the Radiology department at SASH. Clinical baseline characteristics and, when available, additional standard ankle ABPI (*n* = 129) and CT angiogram (CTA; *n* = 120) were also included.

On the first dataset, we performed regression analyses to determine which of the handheld Doppler waveform characteristics best reflected standard ABPI values and to establish best-fit line equations. In the second group of patients, we validated this approach by comparing AccI-derived eABPI with standard ABPI.

In the third dataset, we identified legs (*n* = 100) with ABPI ≤ 1.1, aiming to minimise the contribution of media sclerosis [13], and used these to establish a transfer function for estimating ABPI from angle-corrected Duplex ultrasound images. We then used this transfer function to estimate ABPI (eABPI) in the entire group of 148 legs. We compared the performance of eABPI with standard ABPI to diagnose PAD, using duplex ultrasonography as the reference standard. We also conducted a subgroup analysis to compare performance in the presence or absence of diabetes. Finally, in the subgroup for whom CTA images were available, we compared the diagnostic performance of eABPI with CTA as the reference standard. Note that limbs with previous revascularisation were not included in the diagnostic performance analyses (*n* = 9).

### 2.1. Measurement of Standard ABPI

After 10 min rest in a supine position, systolic blood pressure was determined at each of the brachial, posterior tibial and dorsalis pedis arteries, using a handheld Doppler probe (Huntleigh) and a sphygmomanometer with appropriately sized cuffs placed around the upper arm and ankle. For each leg, the ABPI was calculated by dividing the highest ankle pressure (from either the dorsalis pedis artery or posterior tibial artery) by the highest arm pressure reading.

### 2.2. Doppler Waveform Analysis

In the first two datasets, peripheral arterial Doppler waveforms were assessed using a Huntleigh Mini Dopplex Hand-Held Doppler with 8 Mhz probe. Analogue waveform traces were recorded onto the Huntleigh Dopplex DR4 reporter software for analysis. Peak Systolic Amplitude (peak systolic velocity corrected for baseline at the foot of the systolic rise) and Rise Time (the interval between onset and peak of the systolic Doppler signal) were determined manually, and AccI was calculated by dividing the peak systolic amplitude by the rise time. In the third dataset, AccI was determined automatically by onboard software on angle-corrected duplex images (GE Logiq E9, 9 MHz linear array transducer) of the dorsalis pedis at the proximal foot and posterior tibial artery below the ankle. The vascular scientist performing the Duplex scans was not aware of standard ABPI measurements. Of note, Doppler waveforms were taken at a non-stenotic site, distal to any crural vessel disease identified during the Duplex scan, as the localised Venturi effect on velocity and waveform gradient at the site of the stenosis would generate a falsely elevated eABPI.

### 2.3. Reference Duplex Vascular Ultrasound of Lower Extremity Arteries and CT Angiography

In the third dataset, all patients received a full duplex ultrasound scan of the lower extremities (GE Logiq E9, 9 MHz linear array transducer). All individuals were scanned on the iliac arteries (if possible), femoropopliteal arteries and crural arteries (anterior tibial, posterior tibial and peroneal artery). The most distal part of the anterior tibial artery was the dorsalis pedis artery at the proximal dorsum of the foot and the posterior tibial artery below the medial ankle. A diagnosis of PAD on ultrasound was confirmed when a single or multiple stenoses > 50% (peak systolic velocity ratio > 2.4) or occlusion of the investigated blood vessels was found.

In a subset of patients, routine care involved CT angiograms which were read and reported by staff radiologists, including one of the authors (AP). Similar to Duplex vascular ultrasound, PAD was diagnosed when single or multiple stenoses > 50% or occlusion of the investigated blood vessels were observed by the reporting radiologist.

### 2.4. Statistical Analyses

Data are presented as mean values and standard deviation. Receiver Operating Characteristic Area Under the Curve (ROC AUC) was obtained. Linear logarithmic regression analyses (best-fit curve estimate) were performed for handheld Doppler Rise Time data and non-linear regression analyses for Peak Amplitude and AccI to ensure curve intersection at the origin. All analyses were performed using SPSS 28 (IBM, Armonk, NY, USA).

## 3. Results

### 3.1. Association of Handheld Doppler Wave Form Characteristics with Standard ABPI

Doppler waveform characteristics (peak, rise time, acceleration index [AccI]) and handheld Doppler ABPI were analysed in 100 legs. Regression analysis showed that Peak and AccI were positively, and Rise Time inversely logarithmically associated with ABPI (*R*^2^ = 0.46, 0.68, 0.42–0.44). The AccI showed the strongest association (*p* < 0.001, Figure 1). The best-fit curve had the following empirical equation (E[y] = 0.32 ∗ ln[1.71 ∗ x + 1]), which allows the calculation of y (eABPI) by inserting measured AccI as x.

### 3.2. Validation of ABPI Estimation with Handheld Doppler

Based on the best fit curve established above, we then estimated ABPI based on the AccI of another *n* = 100 legs. ABPI and eABPI are significantly very strongly correlated (Figure 2A; *r* = 0.81, *p* < 0.001). When comparing measured standard ABPI and estimated ABPI, the two measurements differed on average by −0.01 ± 0.13 (SD, see Figure 2B Bland Altman plot).

### 3.3. Establishment of Equation to Calculate eABPI from Angle Corrected Duplex Doppler AccI at the Ankle

We then considered the relationship between AccI on angle-corrected duplex Doppler images and standard ABPI in 100 legs of patients with ABPI ≤ 1.1 aiming to exclude media sclerosis or at least minimise the contribution of media sclerosis. Using a non-linear template function of the form y = A ∗ ln(B ∗ x + 1), which goes through the origin as would theoretically be expected of this relationship, the best fit curve estimate (*R^2^* = 0.92, *p* = 0.006) is given by E[y] = 0.297 ∗ ln(0.039 ∗ x + 1). We then calculated eABPI based on this (Figure 3A). The calculated eABPI and measured ABPI again very strongly correlated (Figure 3B; *r* = 0.96, *p* < 0.001). When comparing measured ABPI and eABPI, the two measurements differed on average by 0.00 ± 0.08 (SD, see Figure 3C Bland Altman plot). An excel based calculator is provided for download alongside this article (see Appendix A).

### 3.4. eABPI Performs Better Than Standard Doppler ABPI to Diagnose PAD Independent of Diabetes

We then tested the performance of eABPI based on angle-corrected AccI in the larger dataset, including patients with diabetes, chronic limb-threatening ischaemia and media sclerosis (*n* = 148 legs, *n* = 85 patients). See Table 1 for clinical characteristics. Figure 4 shows a scatter plot of AccI and standard ABPI with symbols depicting patients without and with diabetes (A) and duplex ultrasound-confirmed PAD (B). On visual inspection, media sclerosis, as defined by standard ABPI > 1.1, was not confined to people with diabetes, and, in this group, eABPI was lower in, and thus more sensitive for, PAD.

We then compared the performance of eABPI and ABPI to diagnose PAD (diagnosis based on Duplex ultrasonography) using ROC analysis, including the legs of patients in which both eABPI and standard ABPI were present. For both eABPI and standard ABPI, the higher ankle value of dorsalis pedis and posterior tibial artery was used, and PAD patients with recent angioplasty were excluded from this analysis as the procedure often increased ABPI to 0.9 or above (*n* = 120 included). The ROC analysis showed that both standard ABPI (Area under the curve [AUC] = 0.91 ± 0.03 (95% CI: 0.86, 0.96) and eABPI (AUC = 0.99 ± 0.01 (95% CI: 0.98, 1.01) performed well as markers of PAD (both *p* < 0.001, Figure 5A) with eABPI being significantly better. Using a cut-off of ≤0.9, eABPI had a higher sensitivity of 97% (vs. 86% standard ABPI) with a specificity of 96% (vs. 100% standard ABPI).

In a subset of the patients (*n* = 120), both CTA and eABPI were available, so we performed a sensitivity analysis, testing the performance of eABPI to diagnose PAD based on CTA as the reference standard. As shown in Figure 5B, this confirmed the results with the vascular ultrasound image standard. The ROC AUC was 0.98 (*p* < 0.001), and a cut-off of ≤0.9 had a sensitivity of 97% and specificity of 100%.

We then performed another sensitivity analysis comparing the diagnostic performance of eABPI and standard ABPI between the legs of patients without and with diabetes (Table 2). In these sub-groups, eABPI with cut-off ≤0.9 had a specificity of 97% in patients without diabetes mellitus and 95% in patients with diabetes mellitus and sensitivity of 95% and 100%, respectively (Figure 6). This indicated no significant overall difference in performance between the two subgroups (*p* = 0.608). The performance of standard ABPI tended to be lower in diabetes than in its absence, with lower sensitivity (85% vs. 89%) without reaching the 0.05 threshold defined for statistical significance (*p* = 0.362).

## 4. Discussion

The key findings of the present study are that the Doppler AccI measured either by a handheld Doppler or Duplex ultrasound of the ankle arteries shows a very strong logarithmic association with standard ABPI. An estimated ABPI can be calculated based on empirical best-fit equations, and a calculator is provided in this publication. The performance of eABPI was better than standard ABPI to diagnose PAD, with eABPI having significantly larger ROC AUC and showing higher sensitivity without relevant loss of specificity independent of diabetes status.

Most current PAD clinical practice guidelines recommend ABPI as a first-line diagnostic test [8,14,15,16,17], with Doppler waveform analysis being used in instances of discrepancy with the clinical picture. However, no specific parameters are given to quantify the Doppler waveform other than that the absence of a multiphasic signal indicates proximal vascular stenoses [11]. The physiological explanations include early reflection of the pulse wave by proximal stenosis, preventing the typical dip which is otherwise caused by the distal reflection of the pressure wave. In addition, increased diastolic velocity indicates low resistance, i.e., dilation of downstream microcirculation occurring through ischaemia. However, increased diastolic flow is also observed during and after exercise in the presence of distal inflammation and/or arteriovenous fistulas.

We have shown a close logarithmic relationship between the slope of the systolic upstroke, the AccI, and the measured ABPI. Only a few papers have investigated Doppler waveforms in the context of PAD detection [12,18]. Trihan et al. reported a moderate (*r* = −0.51) inverse linear correlation of acceleration time as measured in pedal arteries with ABPI and toe brachial index [12]. Our current data (Figure 1B) confirm a logarithmic inverse association between Rise Time and ABPI. However, our data take this one step further. We compared Peak Velocity, Rise Time and AccI and showed that the association between AccI and ABPI is significantly better than for Rise Time or Peak Velocity. We were able to estimate ABPI based on AccI with high precision (average deviation both −0.01). We validated this approach in two independent datasets. As expected, the precision to predict ABPI was higher using angle-corrected (SD 0.08) vs. not-angle-corrected handheld Doppler values (SD 0.13).

NICE guidance on PAD diagnosis [9] acknowledges the fallibility of ABPI in people with diabetes. This arises from a high prevalence of media sclerosis, which invalidates ABPI by violating the essential assumption underlying ABPI measurement—the compressibility of arteries. Our results confirm that the presence of diabetes decreases the performance of standard ABPI (Figure 6). Furthermore, we show that eABPI detects the presence of PAD more accurately than standard ABPI and, importantly, does not require any compression of arteries. In addition, the performance of eABPI was similar regardless of diabetes status.

### 4.1. Limitations

Some limitations apply to the current data. The acquisition of Doppler waveforms with both handheld Doppler and duplex ultrasound was performed by highly trained vascular scientists. Such staff is not widely accessible in primary care, where a simple, reliable means to diagnose PAD is urgently needed. However, our data provide the first proof of concept that ABPI can be estimated precisely and PAD diagnosed accurately, using a single parameter—AccI—from Doppler waveform analysis. As our dataset is only from two sites and was part of service evaluations, further validation, including haemodynamic validation and performance in larger groups at other sites, may be needed to confirm our data. Another important limitation applies to the approach of using AccI. If arterial stenosis is present at the site of measurements, the intra-stenotic flow velocity increase may yield false negative results. Moreover, like standard ABPI, stenoses that are distal to the site of measurement are not detected. Note that the only misclassified leg (eABPI > 0.9 but PAD present) in Figure 6 had a solitary focal stenosis in the distal tibial artery, which could explain the false classification. Finally, a general limitation applies to all ankle-based methodologies as opposed to distal, e.g., toe pressure measurements to evaluate PAD, as they may miss below the ankle disease in diabetic patients [19]. However, in how far the detection of isolated below-the-ankle disease will change clinical decisions in view of established indications for best medical management in diabetic and ulcer patients [8], unclear cost-effectiveness of toe pressure measurements [20] and the unclear effectiveness of plantar arch revascularisation remains to be determined.

### 4.2. Clinical Implications

For the time being, our approach could immediately be implemented into the diagnostic workflow of vascular laboratories and potentially emergency departments to create validation datasets in individual centres using our provided eABPI estimator. We believe that more convenient software-based solutions could be integrated into ultrasound machines and even handheld Doppler devices, making eABPI part of every standard vascular ultrasound exam. In many cases, if a normal eABPI is established as a first step, a full vascular ultrasound exam may not be required, saving valuable resources in the healthcare system. Furthermore, this approach could likely be extended to below the ankle arteries [12] or even toe arteries promising future development to estimate the toe brachial index. With a wider perspective, we believe that our findings open the door to the development of technological solutions for acquiring waveforms on distal arteries in a convenient, inexpensive, user-independent way that would streamline diagnosis and address healthcare-access inequalities.

## 5. Conclusions

This is the first study to demonstrate that eABPI calculation based on the Doppler AccI, without arterial compression, diagnoses PAD more reliably than standard ABPI. eABPI detects PAD in patients with media sclerosis, where ABPI fails. This simple approach can be readily implemented into clinical practice using the calculator application provided as part of this publication.

## Figures and Tables

**Figure 1 jcm-12-00097-f001:**
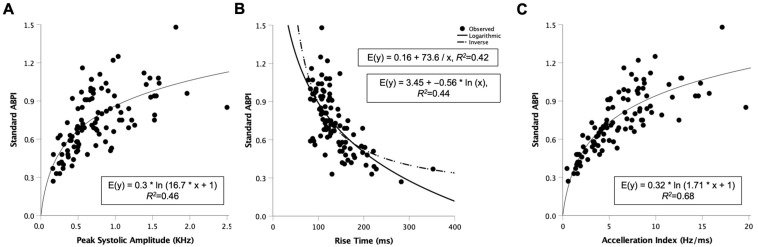
Regression analysis to find best-fit curve estimates to predict/estimate ankle brachial pressure index (ABPI). (**A**) Peak Systolic Amplitude of handheld Doppler at the ankle artery without angle correction, reflecting peak systolic velocity, (**B**) interval between onset and peak of systolic Doppler signal (Rise Time) and (**C**) AccI reflecting the slope of the straight line between the foot and peak of the systolic curve. Each parameter was compared with standard ABPI measurements using an ankle blood pressure cuff. AccI was the strongest predictor with R^2^ of 0.68, indicating that 68% of variability is explained (*n* = 100).

**Figure 2 jcm-12-00097-f002:**
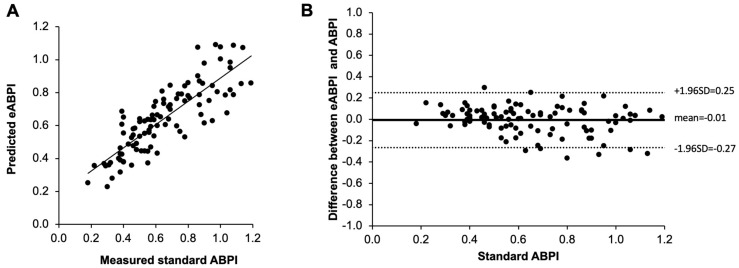
Validation of eABPI using AccI from handheld Doppler and best-fit curve estimate from Figure 1C compared to standard ABPI measured with handheld Doppler probe and cuff occlusion. (**A**) Significant correlation between eABPI and ABPI (*r* = 0.85, *p* < 0.001). (**B**) Bland Altman plot demonstrating an average deviation of −0.01 when comparing eABPI with ABPI with a standard deviation of differences of 0.13 (*n* = 100).

**Figure 3 jcm-12-00097-f003:**
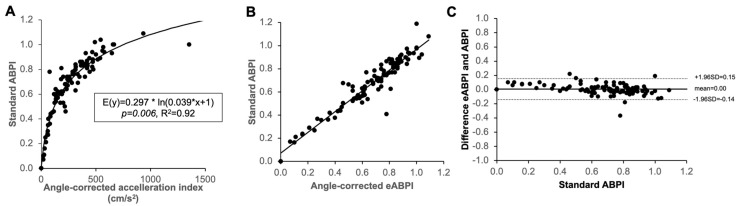
(**A**) Non-linear regression analysis to derive best-fit curve for eABPI using AccI from angle-corrected duplex Doppler, comparing with standard ABPI in limbs aiming to exclude media sclerosis (ABPI ≤ 1.1). (**B**) Significant correlation between eABPI and standard ABPI (r = 0.96, *p* < 0.001). (**C**) Bland Altman plot demonstrating an average deviation of 0.00 when comparing eABPI with standard ABPI with a standard deviation of differences of 0.08 (*n* = 100).

**Figure 4 jcm-12-00097-f004:**
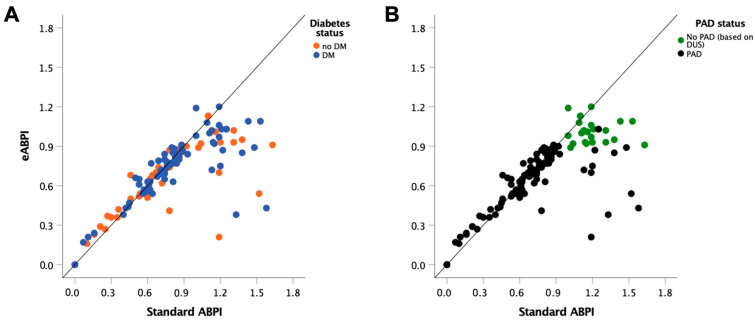
Scatter plots showing individual data of *n* = 129 legs with standard ABPI vs. angle-corrected eABPI in patients without previous angioplasty. Individual data points labelled according to (**A**) diabetes mellitus (DM, *n* = 78, 56%) status and (**B**) presence of peripheral artery disease (PAD, *n* = 102, 73%) according to duplex ultrasound (DUS) imaging reference.

**Figure 5 jcm-12-00097-f005:**
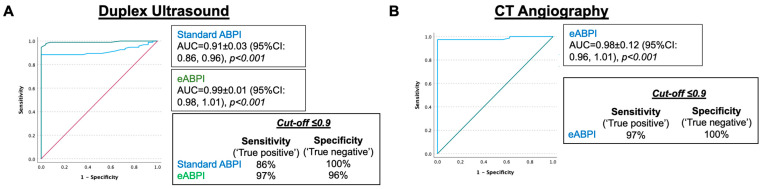
ROC analysis to test the diagnostic performance of eABPI (**A**,**B**) and standard ABPI (**A**) to detect PAD as (**A**) diagnosed with Duplex ultrasound (*n* = 117) and (**B**) computed tomography (CT) angiography (*n* = 111) as imaging standards.

**Figure 6 jcm-12-00097-f006:**
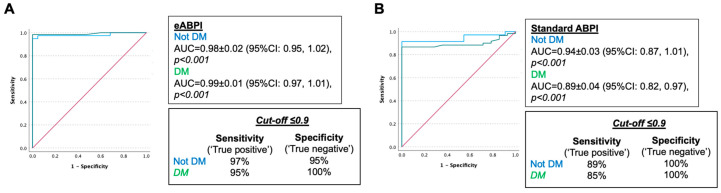
ROC analysis to test the performance of (**A**) eABPI and (**B**) standard ABPI to detect PAD (diagnosed as > 50% stenosis on Duplex ultrasound) in legs of patients without and with diabetes mellitus (DM). (**A**) total *n* = 139 legs, *n* = 78 DM, *p* = 0.608, (**B**) total = 120 legs, *n* = 74 DM, *p* = 0.349.

**Table 1 jcm-12-00097-t001:** Characteristics of patients in study 3. (mean±standard deviation; ABPI = ankle brachial pressure index, CTA = computed tomography angiography, eABPI = estimated ankle brachial pressure index, PAD = peripheral artery disease).

Limbs (*n*)	148	
Patients (*n*)		85
Vascular ultrasound (*n*)	148	85
CTA (*n*)	120 (81%)	57 (67%)
Standard ABPI (*n*)	129 (87%)	77 (91%)
Standard ABPI ≤1.1 (*n*)	100 (68%)	60 (78%)
PAD based on ultrasound (*n*)	111 (75%)	65 (76%)
PAD based on CTA (*n*)	86 (72%)	48 (56%)
Rutherford 0–2 (*n*)	6 (4%)	2 (2%)
Rutherford 3 (*n*)	24 (16%)	16 (19%)
Rutherford 4 (*n*)	13 (9%)	6 (7%)
Rutherford 5 (*n*)	64 (43%)	39 (46%)
Chronic limb-threatening ischaemia (*n*)	67 (52%)	45 (53%)
Post angioplasty (*n*)	9 (6%)	2 (2%)
Age (years)	70 ± 11	70 ± 12
Diabetes mellitus (*n*)	76 (51%)	45 (53%)
Type 1 (*n*)	8 (5%)	4 (5%)
Type 2 (*n*)	65 (44%)	39 (46%)
Unknown type (*n*)	3 (2%)	2 (2%)
Arterial hypertension (*n*)	109 (74%)	63 (74%)
Chronic obstructive pulmonary disease (*n*)	32 (22%)	17 (20%)
Coronary artery disease (*n*)	50 (34%)	30 (35%)
Chronic heart failure (*n*)	30 (20%)	18 (21%)
Chronic kidney disease (*n*)	72 (49%)	42 (49%)
Stroke (*n*)	16 (11%)	10 (12%)
Cancer (*n*)	15 (10%)	9 (11%)
Abdominal aortic aneurysm (*n*)	3 (2%)	2 (2%)
Atrial fibrillation (*n*)	27 (18%)	16 (19%)
Smoker (*n*)	29 (20%)	17 (20%)
Ex-Smoker (*n*)	85 (57%)	51 (60%)

**Table 2 jcm-12-00097-t002:** Characteristics of patients with and without diabetes mellitus. (mean ± standard deviation, independent *t*-test, Characteristics are described according to limbs; ABPI = ankle brachial pressure index, eABPI = estimated ankle brachial pressure index AccI = Acceleration Index, CT = computed tomography, eABPI = estimated ankle brachial pressure index, PAD = peripheral artery disease).

	Not Diabetes Mellitus	Diabetes Mellitus	*p*
Limbs (*n*)	65	83	
Patients (*n*)	40	45	
Vascular ultrasound (*n*)	65 (100%)	83 (100%)	
CT angiogram (*n*)	49 (75%)	53 (64%)	
Standard ABPI (*n*)	50 (77%)	79 (95%)	
Standard ABPI	0.77 ± 0.38	0.83 ± 0.34	0.225
AccI (cm/s^2^)	419 ± 321	384 ± 291	0.113
eABPI	0.74 ± 0.29	0.74 ± 0.25	0.078
PAD based on ultrasound (*n*)	43 (66%)	68 (82%)	
PAD based on CTA (*n*)	36 (73%)	50 (94%)	
Rutherford 0–2 (*n*)	3 (4%)	3 (4%)	
Rutherford 3 (*n*)	17 (20%)	7 (8%)	
Rutherford 4 (*n*)	11 (13%)	2 (2%)	
Rutherford 5 (*n*)	12 (14%)	52 (63)	
Chronic limb-threatening ischaemia (*n*)	23 (28%)	54 (65%)	
Post angioplasty (*n*)	4 (5%)	5 (6%)	
Age (years)	71 ± 10	69 ± 12	0.289
Arterial hypertension (*n*)	42 (51%)	67 (81%)	
Chronic pulmonary disease (*n*)	13 (16%)	19 (23%)	
Coronary artery disease (*n*)	14 (17%)	36 (43%)	
Chronic heart failure (*n*)	9 (11%)	21 (25%)	
Chronic kidney disease (*n*)	24 (19%)	48 (58%)	
Stroke (*n*)	4 (5%)	12 (14%)	
Cancer (*n*)	7 (8%)	8 (10%)	
Abdominal aortic aneurysm (*n*)	2 (2%)	1 (1%)	
Atrial fibrillation (*n*)	15 (18%)	12 (14%)	
Smoker (*n*)	15 (18%)	14 (17%)	
Ex-Smoker (*n*)	33 (40%)	52 (63%)	

## Data Availability

As this was part of an NHS service evaluation, the data are not publicly available.

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
