# Peer review of "Ankle Doppler for Cuffless Ankle Brachial Index Estimation and Peripheral Artery Disease Diagnosis Independent of Diabetes"

_jcm, 2022, doi:10.3390/jcm12010097_

Round 1
Reviewer 1 Report
This work proposes a new modality for a simple assessment of PAD, especially in diabetic patients and those with calcified arteries, in whom the ABI is not reliable.
The data analysis is accurate, and variability has been assessed. Limitations and future perspectives are well expressed. This work provides an exciting starting point to further examine the use of Accl detection in clinical practices. It is a definitely proof of concept.
·
Author Response
We thank the reviewer for his/her time to review the manuscript and kind acknowledgement of our work.
Reviewer 2 Report
The authors present in their manuscript a novel method to estimate the ABI based on an ankle Doppler acceleration index from retrospective data of three patient cohorts.
I recommend to adapt the title - the abbrevation PAD in the title is not ideal - please use "peripheral arterial disease" instead
Abstract and introduction are well written and give a good overview.
What was the inclusion period of the study? Were these consecutive patients or some kind of selected?
In line 86-87 the authors state "we identified legs (n=100) without media sclerosis, based on 86 standard ABPI ≤1.1" - the reference 13 as all following references are not listed in the manuscript.
Also, as the authors stated before the ABPI below 1.1 is not a guarantee that there is no media sclerosis in these patients. (this is again mentioned in line 166-167).
Table 1 has several inconsistencies:
- First it is strange that only 148 limbs were scanned in 84 patients. How many amputees were in this cohort?
- Why are several percentages not listed?
- Is it really approriate to list that 51% of limbs had diabetes instead of the percentage of patients? (same goes for cancer etc.).
In Figure legend 4 71% instead of 70% of patients had PAD.
In Table 2 I recommend to use the Rutherford classification instead of Fontaine.
In line 233-234 the authors state "The performance of eABPI was better than standard ABPI to diagnose PAD independent of diabetes status." - I recommend to adapt this to something more specific and scientific - e.g. "eABPI showed higher sensitivity without relevant loss of specificity"
- Again reference #13 to 19 are not listed.
In the limitations section about 50% of the text are not limitations and should be part of the discussion. e.g. lines 285-289.
In the conclusion the reference to the application in the supplementary files is not ideal - I would rather move this to the methods section or the discussion.
The authors regularly mention the limitations of standard ABPI - what is the role of toe pressure or a toe brachial index in their practice? It is commonly used, especially in diabetic patients with media sclerosis.
Author Response
The authors present in their manuscript a novel method to estimate the ABI based on an ankle Doppler acceleration index from retrospective data of three patient cohorts.
I recommend to adapt the title - the abbrevation PAD in the title is not ideal - please use "peripheral arterial disease" instead
Done
Abstract and introduction are well written and give a good overview.
Thanks.
What was the inclusion period of the study? Were these consecutive patients or some kind of selected?
We have added the inclusion period. Patients were consecutive patients as stated in line 69.
In line 86-87 the authors state "we identified legs (n=100) without media sclerosis, based on 86 standard ABPI ≤1.1" - the reference 13 as all following references are not listed in the manuscript.
Sorry for this. References were added.
Also, as the authors stated before the ABPI below 1.1 is not a guarantee that there is no media sclerosis in these patients. (this is again mentioned in line 166-167).
We understand this point in a sense that we should make more clear that ABPI < 1.1 is no guarantee that media sclerosis is excluded. We have reworded the respective sections: “aiming to minimise the contribution of media sclerosis” and “with ABPI £1.1 aiming to exclude media sclerosis or at least minimise the contribution of media sclerosis”
Table 1 has several inconsistencies:
- First it is strange that only 148 limbs were scanned in 84 patients. How many amputees were in this cohort?
None of the patients had a major amputation. However, the data were gathered as a prospective clinical service evaluation and the sonographers in the radiology department did not always perform exams on both legs. Due to shortage of staff they sometimes only focused on a symptomatic limb.
- Why are several percentages not listed?
Percentages added to both tables.
- Is it really approriate to list that 51% of limbs had diabetes instead of the percentage of patients? (same goes for cancer etc.).
Whereas we believe this is the appropriate approach as this is a per limb analysis, we have added characteristics per patient to table 1. The distribution is very similar when assessing on a per patient basis. We have however elected not to add patient data to table 2 as it would make the table to crowded and does not provide critical information as all analyses are per limb.
In Figure legend 4 71% instead of 70% of patients had PAD.
We have corrected the figure legend. In fact, standard ABPI was only available in 129 and 102 of the 129 had confirmed PAD (73%). Note that these are data without previous angioplasty which explains the (on first sight) discrepant number as compared to table 1. We have corrected the error and stated this in the legend.
In Table 2 I recommend to use the Rutherford classification instead of Fontaine.
Exchanged.
In line 233-234 the authors state "The performance of eABPI was better than standard ABPI to diagnose PAD independent of diabetes status." - I recommend to adapt this to something more specific and scientific - e.g. "eABPI showed higher sensitivity without relevant loss of specificity"
The ROC analysis showed that the area under curve was statistically significantly bigger indicating better performance. The statistical test does not test if sensitivity or specificity are significantly different. We have,however revised this as per the suggestion.
- Again reference #13 to 19 are not listed.
References added. Sorry, likely a technical issue.
In the limitations section about 50% of the text are not limitations and should be part of the discussion. e.g. lines 285-289.
We agree and have move large parts into a new section entitle “clinical implications”.
In the conclusion the reference to the application in the supplementary files is not ideal - I would rather move this to the methods section or the discussion.
The supplementary file is an automated calculator. We agree that this is not best placed in the conclusion of the manuscript. We believe it suffices to state that the calculator is provided as part of the publication (which is an important added value to the publication as it allows immediate implementation by others). We have also added this statement to the results section.
The authors regularly mention the limitations of standard ABPI - what is the role of toe pressure or a toe brachial index in their practice? It is commonly used, especially in diabetic patients with media sclerosis.
In our practise, which has a great proportion of diabetic foot ulcer patients with a large interventional focus, TBI is used where the results of which are thought to be likely to change clinical decisions and in terms of procedure planning. An example would be in cases where the ABPI/eABPI is normal despite clinical suspicion of ischaemia and (relative) contraindications for intra-arterial angiography. We find it is also of benefit in patients with decreased ABPI/eABPI and complex revascularisation options who require digital amputations – TBI provides information of whether such amputations are likely to heal without revascualrisation In our experience TBI only rarely changes the clinical decisions but rather may give the interventionalist or patient assurance that an angiography or complex revascularisation is indeed needed if any doubts exist.
We have added a section to the limitations section: “Finally, a general limitation applies to all ankle based methodologies as opposed to distal toe pressure measurements to evaluate PAD as they may miss below the ankle disease in diabetic patients. However, in how far the detection of isolated below the ankle disease will change clinical decisions in view of established indication for best medical management in diabetic and ulcer patients (8), unclear cost-effectiveness of toe pressure measurements and the unclear effectiveness of plantar arch revascularisation remains to be determined.”